# TcpC inhibits toll-like receptor signaling pathway by serving as an E3 ubiquitin ligase that promotes degradation of myeloid differentiation factor 88

Jia-qi Fang[1,2‡], Qian Ou[1,2‡], Jun Pan[3‡], Jie Fang[1,4], Da-yong Zhang[1,4], Miao-qi Qiu[4], Yue-qi Li[4], Xiao-Hui Wang[4], Xue-yu Yang[4], Zhe Chi[1,2], Wei Gao[1,4], Jun-ping Guo[1,4], Thomas Miethke[5]*, Jian-ping Pan[1,4]*

1 Institute of Translational Medicine, Zhejiang University City College, Hangzhou, Peoples Republic of China, 2 Institute of Immunology, Zhejiang University School of Medicine, Hangzhou, Peoples Republic of China, 3 Cancer Institute, Second Affiliated Hospital, Zhejiang University School of Medicine, Hangzhou, Peoples Republic of China, 4 Department of Clinical Medicine, Zhejiang University City College School of Medicine, Hangzhou, Peoples Republic of China, 5 Institute of Medical Microbiology and Hygiene, Medical Faculty of Mannheim, University of Heidelberg, Mannheim, Germany

‡ These authors share first authorship on this work.
* Thomas.Miethke@medma.uni-heidelberg.de (TM); jppan@zucc.edu.cn (JP)

**Data Availability Statement:** All relevant data are within the manuscript and its Supporting Information files.

## Abstract

TcpC is a virulence factor of uropathogenic *E. coli* (UPEC). It was found that TIR domain of TcpC impedes TLR signaling by direct association with MyD88. It has been a long-standing question whether bacterial pathogens have evolved a mechanism to manipulate MyD88 degradation by ubiquitin-proteasome pathway. Here, we show that TcpC is a MyD88-targeted E3 ubiquitin ligase. Kidney macrophages from mice with pyelonephritis induced by TcpC-secreting UPEC showed significantly decreased MyD88 protein levels. Recombinant TcpC (rTcpC) dose-dependently inhibited protein but not mRNA levels of MyD88 in macrophages. Moreover, rTcpC significantly promoted MyD88 ubiquitination and accumulation in proteasomes in macrophages. Cys12 and Trp106 in TcpC are crucial amino acids in maintaining its E3 activity. Therefore, TcpC blocks TLR signaling pathway by degradation of MyD88 through ubiquitin-proteasome system. Our findings provide not only a novel biochemical mechanism underlying TcpC-medicated immune evasion, but also the first example that bacterial pathogens inhibit MyD88-mediated signaling pathway by virulence factors that function as E3 ubiquitin ligase.

## Author summary

Toll/interleukin-1 receptor domain-containing protein encoded by *E. coli* (TcpC) is an important virulence factor in many strains of uropathogenic *E. coli* (UPEC). TcpC-mediated evasion of innate immunity plays an important role in the pathogenesis of UPEC caused urinary tract infection (UTI) including pyelonephritis. In the present study, we show TcpC is an E3 ubiquitin ligase that promotes ubiquitination and degradation of

**Funding:** This work was supported by grants from National Natural Science Foundation of China (81671613 and 30872325 to J.P.P.), the Natural Science Foundation of Zhejiang Province (LQ20H100001 to J.Q.F.), the Postdoctoral Science Foundation of China (2020M671747 to J.Q.F) and the Science and Technology Bureau of Zhejiang Province (LGD20H100001 to J.F.). The funders had no role in study design, data collection and analysis, decision to publish, or preparation of the manuscript.

**Competing interests:** The authors have declared that no competing interests exist.

MyD88, hereby blocking the TLR signaling pathway. Our findings not only illuminate the novel biochemical mechanisms underlying TcpC-mediated evasion of innate immunity, but also provide the first example that bacterial pathogens can subvert TLR signaling pathway through virulence factors that function as MyD88-targeted E3 ubiquitin ligase.

## Introduction

Urinary tract infection (UTI) is caused by various pathogens and in patients with pyelonephritis can lead to renal failure [1–3]. The disease is affecting 150 million people worldwide each year and the medical expenses exceed 6 billion dollars [4–6]. Uropathogenic *Escherichia coli* (UPEC) can specifically adhere to and infect epithelial cells of urinary tract mucosa [7,8]. UPEC is the most common pathogen of UTI, about 80% of UTI are caused by UPEC [9].

Toll/interleukin-1 receptor (TIR) domain-containing protein encoded by *E. coli* (TcpC) is a crucial virulence factor in many strains of UPEC [10,11]. *tcpc* is widely distributed among clinical extraintestinal pathogenic *Escherichia coli* (ExPEC) isolates with almost half of the *E. coli* strains from patients suffering pyelonephritis shown to be *tcpc* positive as compared to only 8% in commensal isolates [12].

TIR domain of TcpC (TcpC-TIR) impedes toll-like receptor (TLR) signaling pathway by direct association with myeloid differentiation factor 88 (MyD88) and TLR4 through its predicted DD and BB loops, hereby subverting the innate immune response, promoting bacterial survival and increasing the severity of UTIs in human and mice [10,11]. The mortality rate of mice with pyelonephritis induced by TcpC-secreting wild-type UPEC strain CFT073 was significantly increased compared to the rate of mice infected with the TcpC-deficient strain [11,13]. In addition, TcpC can up-regulate claudin-14 and enhance the barrier function of intestinal epithelial cells by activating PKCζ_and ERK1/2-mediated signaling pathways [14]. Finally, TcpC inhibits the activation of the NLRP3 inflammasome by binding to NLRP3 and caspase-1, leading to inhibition of IL-1β production in macrophages [15]. Therefore, TcpC is a multifunctional virulence factor that inhibits innate immunity.

Nuclear magnetic resonance titration experiments have demonstrated that TcpC-TIR interacts with CD, DE and EE loops on the surface of MyD88, suggesting that TcpC specifically engages these MyD88 structural elements for immune suppression [10]. But, it is obvious that inhibition of the TLR signaling pathway just by presumably competitive binding to MyD88 does not represent a very efficient way, since UPECs need to produce large amounts of TcpC to achieve a competitive advantage. It has been a long-standing question if bacterial pathogens have evolved a more efficient mechanism to manipulate MyD88 degradation by post-translational modification. Here we show that TcpC is a MyD88-targeted E3 ubiquitin ligase that promotes degradation of MyD88, the key adaptor protein of the TLR signaling pathway. Our findings not only illuminate the biochemical mechanisms underlying TcpC-medicated immune evasion, but also provide novel clues to clarify the pathogenicity of bacterial pathogens.

## Results

### Kidney macrophages from mice suffering from pyelonephritis induced by TcpC-secreting CFT073 show decreased levels of MyD88 proteins

We infected mice by transurethral instillation of TcpC-secreting wild-type UPEC strain CFT073 (CFT073$^{wt}$) or *tcpc* knock out CFT073 mutant (CFT073$^{\Delta tcpc}$). 7 of 9 mice developed

macroscopic signs of acute pyelonephritis in CFT073[wt] group but none in other groups. Abscesses in kidneys could be seen in CFT073[wt]-induced pyelonephritis mouse model, no abscess was present in kidneys from CFT073[Δtcpc]-infected mice (Fig 1A and 1B). Accordingly,

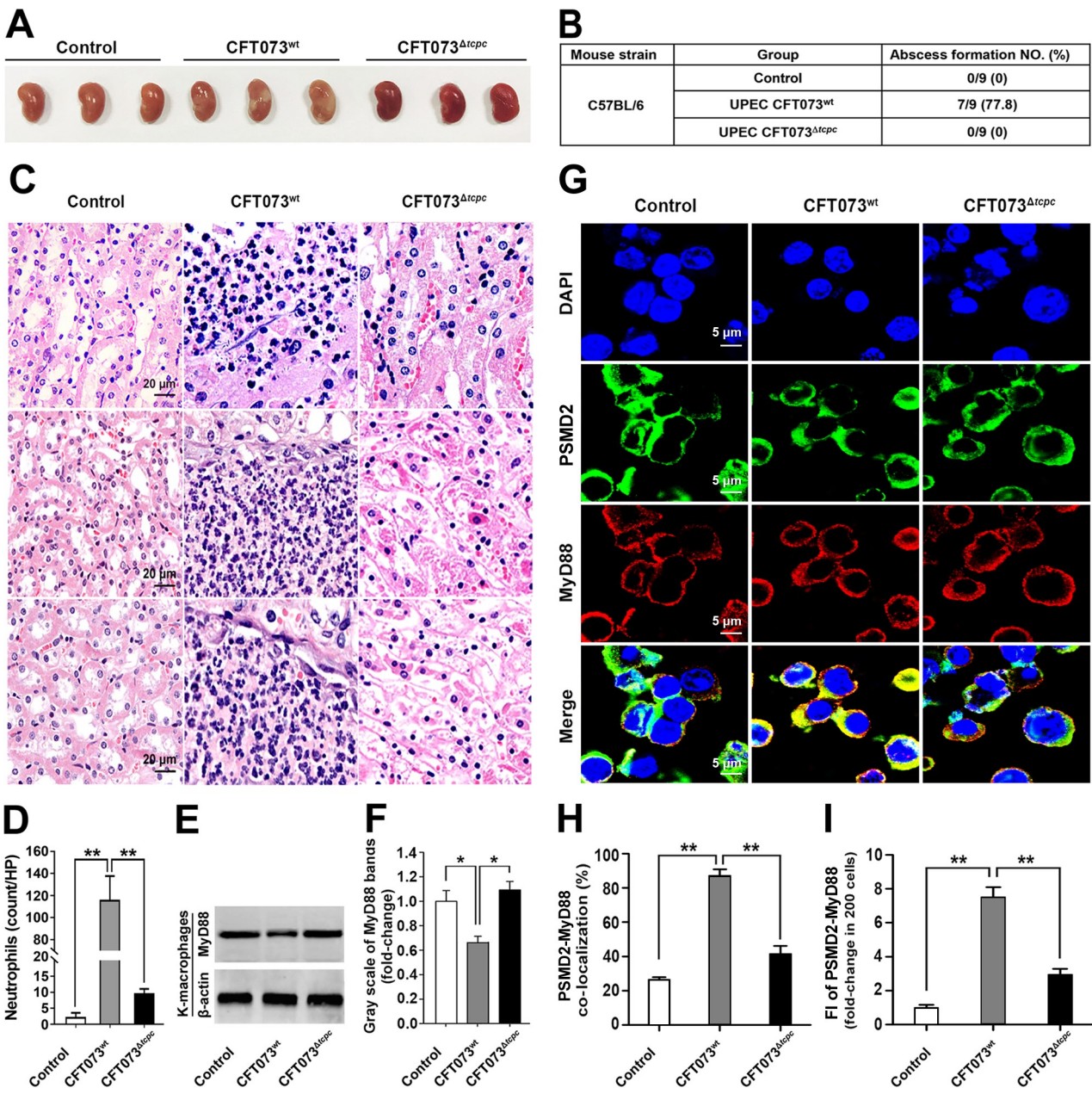

**Fig 1. K-macrophages from TcpC-secreting CFT073 induced pyelonephritis murine model show decreased levels of MyD88 proteins.** (A) Gross pathological changes of kidneys from CFT073[wt]- or CFT073[Δtcpc]-induced murine pyelonephritis models. (B) Percentages of abscess formation in different groups. (C) Histological examinations of kidneys from pyelonephritis mouse models. Note that kidneys from CFT073[wt] infections display a higher level of neutrophil infiltrates. (D) Statistical summary of neutrophil counts in C. Mean ± SD of three independent experiments were shown. **: $p < 0.01$. (E) Western blot analyses of MyD88 protein levels in K-macrophages derived from model mice with pyelonephritis and control mice. (F) Gray scale analyses of bands reflecting MyD88 protein levels in experiments as described in E. The gray scale values of immunoblotting bands from control group were set as 1.0. *: $p < 0.05$. (G) Co-localization of MyD88 with PSMD2, determined by confocal microscopy, in K-macrophages derived from pyelonephritis murine models and from normal saline treated mice (Control). Yellow spots indicate MyD88-PSMD2 co-localization. (H) Statistical summary of MyD88-PSMD2 co-localization percentages. Mean ± SD of three independent experiments were shown. **: $p < 0.01$. (I) Statistical summary of yellow fluorescence intensity (FI) reflecting the MyD88-PSMD2 co-localization in experiments as described in A. The yellow FI values from the control group were set as 1.0. **: $p < 0.01$.

severe neutrophil infiltrates in the kidneys could be observed in the group of CFT073$^{wt}$ induced mice compared to the group infected with CFT073$^{\Delta tcpc}$ (Fig 1C and 1D). Kidney macrophages (K-macrophages) were sorted and the cell purities were >95% when identified by the macrophage marker CD11b and F4/80 (S1A Fig). Interestingly, MyD88 protein levels were decreased (Fig 1E and 1F), while co-localization of MyD88 with the proteasome marker PSMD2, the specific marker for 19S subunit of the proteasome [16], was enhanced significantly in K-macrophages from CFT073$^{wt}$ infected mice with pyelonephritis compared to K-macrophages isolated from mice infected with CFT073$^{\Delta tcpc}$ (Fig 1G–1I). These data suggest that TcpC might promote degradation of MyD88.

## CFT073$^{wt}$ inhibits pro-inflammatory cytokine expression and MyD88 protein level, but shows enhanced survival ability in macrophages

Mouse bone marrow derived macrophages (BMDM) with a purity of about 93% (S1B Fig) and macrophage cell lines (J774A.1 and RAW264.7) were infected by CFT073$^{wt}$ and CFT073$^{\Delta tcpc}$ at a multiplicity of infection (MOI) of 100 for 12 h. The number of intracellular living bacteria in the group of CFT073$^{wt}$ was significantly higher than that in the group of CFT073$^{\Delta tcpc}$ (Fig 2A–2C), suggesting that TcpC favors the survival of UPEC in the macrophages. To examine the influence of TcpC on the expression of cytokines and MyD88 in macrophages, BMDM, J774A.1 and RAW264.7 were separately co-cultured in transwell with CFT073$^{wt}$ and CFT073$^{\Delta tcpc}$ respectively, cytokines and MyD88 levels in the treated macrophages were examined. Our data showed that both the mRNA and protein levels of IL-1β, IL-6 and TNF-α in macrophages treated by CFT073$^{wt}$ were significantly inhibited, when compared with those in CFT073$^{\Delta tcpc}$ treated groups (Fig 2D). MyD88 protein levels were also significantly decreased in macrophages treated with CFT073$^{wt}$ compared with that in macrophages treated with CFT073$^{\Delta tcpc}$ (Fig 2E and 2F). In accordance with the changes in MyD88, levels of p-p50 and p-p65 of NF-κB decreased profoundly in CFT073$^{wt}$ treated BMDM, J774A.1 and RAW264.7 (Fig 2G and 2H), demonstrating that TcpC blocks the TLR signaling pathway.

## rTcpC dose-dependently inhibits MyD88 protein but not mRNA levels in macrophages

To further investigate the mechanisms by which TcpC decreases MyD88 level in macrophages, lipopolyssacharide (LPS)-free recombinant TcpC (rTcpC) was prepared (S2A, S2B and S2D Fig). Mouse macrophage cell lines J774A.1 and RAW264.7 were treated with different concentrations of rTcpC (1, 2, 4 or 8 μg/ml) at 37˚C for the indicated time. Western Blot and qRT-PCR analyses confirmed that rTcpC dose-dependently inhibited MyD88 protein (Fig 3A–3D) but not mRNA (Fig 3E and 3F) levels in J774A.1 and RAW264.7. When human macrophage cell line THP-1 and mouse BMDM were used, results mirrored the same trend were also obtained (S3A–S3F Fig). These data, together with our previous findings that K-macrophages derived from CFT073$^{wt}$ induced pyelonephritis showed enhanced co-localization of MyD88 with proteasome, indicate that TcpC promotes the degradation of MyD88 presumably by affecting the post-translational modification of MyD88.

## rTcpC promotes ubiquitination of MyD88 in macrophages

Ubiquitination plays an important role in protein degradation, stress response, cell cycle regulation, protein trafficking, signal transduction and transcriptional regulation [17–21]. The influence of rTcpC on ubiquitination of MyD88 in macrophages was examined. Proteins that bind to MyD88 in the lysates of J774A.1 and RAW264.7 treated with or without rTcpC were

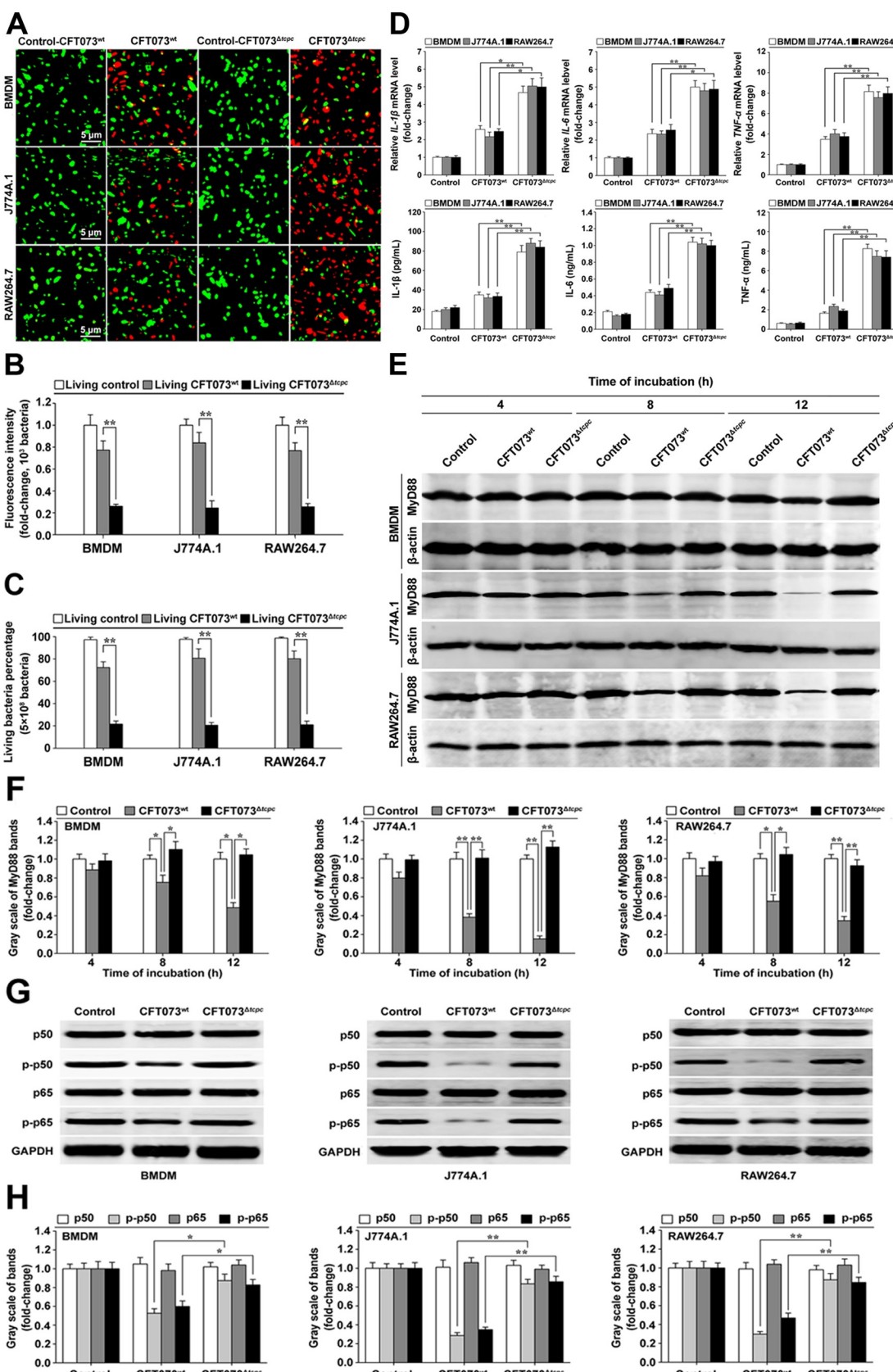

**Fig 2. CFT073^wt inhibits expression of pro-inflammatory cytokines and MyD88, but shows enhanced survival ability in macrophages.** (A) Confocal microscopy examination of survival of CFT073^wt and CFT073^Δtcpc in macrophages. The bacteria stained with SYTO 9 fluorescence dye (green) were living, while the bacteria stained with propidium iodide dye (red) were dead. Scale bar = 5 μm. (B) Quantification of green FI reflecting the living bacteria in macrophages, determined by confocal microscopy. One thousand bacteria in each experiment of three were analyzed to quantify the values of green FI. The green FI value reflecting the living bacteria cultured in RPMI-1640 medium (living control) was set as 1.0. $^{**}$: $p < 0.01$. (C) Living bacteria percentage in macrophages determined by fluorospectrophotometry. Living bacteria in macrophages were examined by fluorescence dye staining as described in A. $5 \times 10^8$ bacteria in each experiment of three were analyzed to quantify the fluorospectrophotometric values. $^{**}$: $p < 0.01$. (D) qRT-PCR and ELISA detection of IL-1β, IL-6 and TNF-α mRNA and protein levels in CFT073^wt- or CFT073^Δtcpc-treated macrophages. $^{*}$: $p < 0.05$, $^{**}$: $p < 0.01$. (E) Dynamic analyses of MyD88 protein level in CFT073^wt- and CFT073^Δtcpc-treated macrophages by Western Blot. (F) Gray scale analyses of MyD88 immunoblotting bands in CFT073^wt- and CFT073^Δtcpc-treated macrophages. The mean ± SD of three independent experiments were shown. The gray scale values of immunoblotting bands from the untreated cells (control) were set as 1.0. $^{*}$: $p < 0.05$; $^{**}$: $p < 0.01$. (G) Western blot analyses of levels of NF-κB p50, p65, p-p-50 and p-p65 in CFT073^wt- and CFT073^Δtcpc-treated macrophages at 16 h. (H) Gray scale analyses of NF-κB p50, p65, p-p-50 and p-p65 immunoblotting bands in three experiments described in G. The gray scale values of immunoblotting bands from the untreated cells (control) were set as 1.0. $^{*}$: $p < 0.05$; $^{**}$: $p < 0.01$.

co-immunoprecipitated by MyD88-IgG, and ubiquitination of MyD88 was determined by immunoblotting. Although protein levels of MyD88 were significantly inhibited in rTcpC treated J774A.1 and RAW264.7, ubiquitination of MyD88 was enhanced profoundly compared with the rTcpC untreated group (Fig 4A–4D). When the proteasome inhibitor MG-132 was employed, the enhanced degradation of MyD88 caused by rTcpC was abrogated, and the ubiquitination of MyD88 was enhanced further compared with the rTcpC treated group (Fig 4A–4D), indicating that rTcpC promotes MyD88 degradation via enhancing ubiquitination of MyD88.

## Bioinformatics analyses show there are WW and PY motifs in TcpC and MyD88 respectively

Ubiquitination is mediated by an enzymatic reaction cascade, a process that proteins are degraded by one or more ubiquitin molecules in combination with ubiquitin-activating (E1), ubiquitin-conjugating (E2), and ubiquitin-ligating (E3) enzymes [22]. E3 not only binds to E2, but also specifically binds to substrate proteins, which plays a key role in the process of ubiquitination [23]. The human genome contains >600 annotated ubiquitin E3 ligases, they can be divided into three types of domains: RING (Really Interesting New Gene), HECT (Homologous to EA6P C-Terminus) and RBR (RING-between-RING) [24]. Previous reports confirmed that a cysteine at the N-terminus of the HECT-like E3 ligase is necessary for its function of ubiquitination [25,26]. HECT-like E3 ligase contains tryptophan-tryptophan (WW) domain and can bind to substrates which have proline-tyrosine (PY) motifs [19]. PY motifs in substrate structure of HECT-like E3 ubiquitin ligase are known to mediate the interaction with WW domains of the HECT-like E3 ubiquitin ligase [27,28]. Our bioinformatics prediction showed that TcpC contains a TIR domain (171–304 aa) and some ubiquitination functional sites (Fig 4E). The Cys residue (C12) and WW domain (W104-W106) are located at the N-terminal sequence of TcpC (Fig 4F), and PY motif is situated on the human and mouse sequence of MyD88 (Fig 4G and 4H). These data imply that TcpC might be an E3 ligase.

## rTcpC is an E3 ubiquitin ligase that enhances ubiquitination of MyD88

To verify whether TcpC acts as an E3 ubiquitin ligase that enhances ubiquitination of MyD88, a ubiquitination assay kit which mimics the ubiquitination process was used as previously described [29,30]. The ubiquitination kit tests demonstrated that rTcpC could replace the E3 ubiquitin ligase for strengthening ubiquitination of both proteins in the lysates of J774A.1 and RAW264.7 and commercially available recombinant MyD88 (rMyD88). Furthermore, this

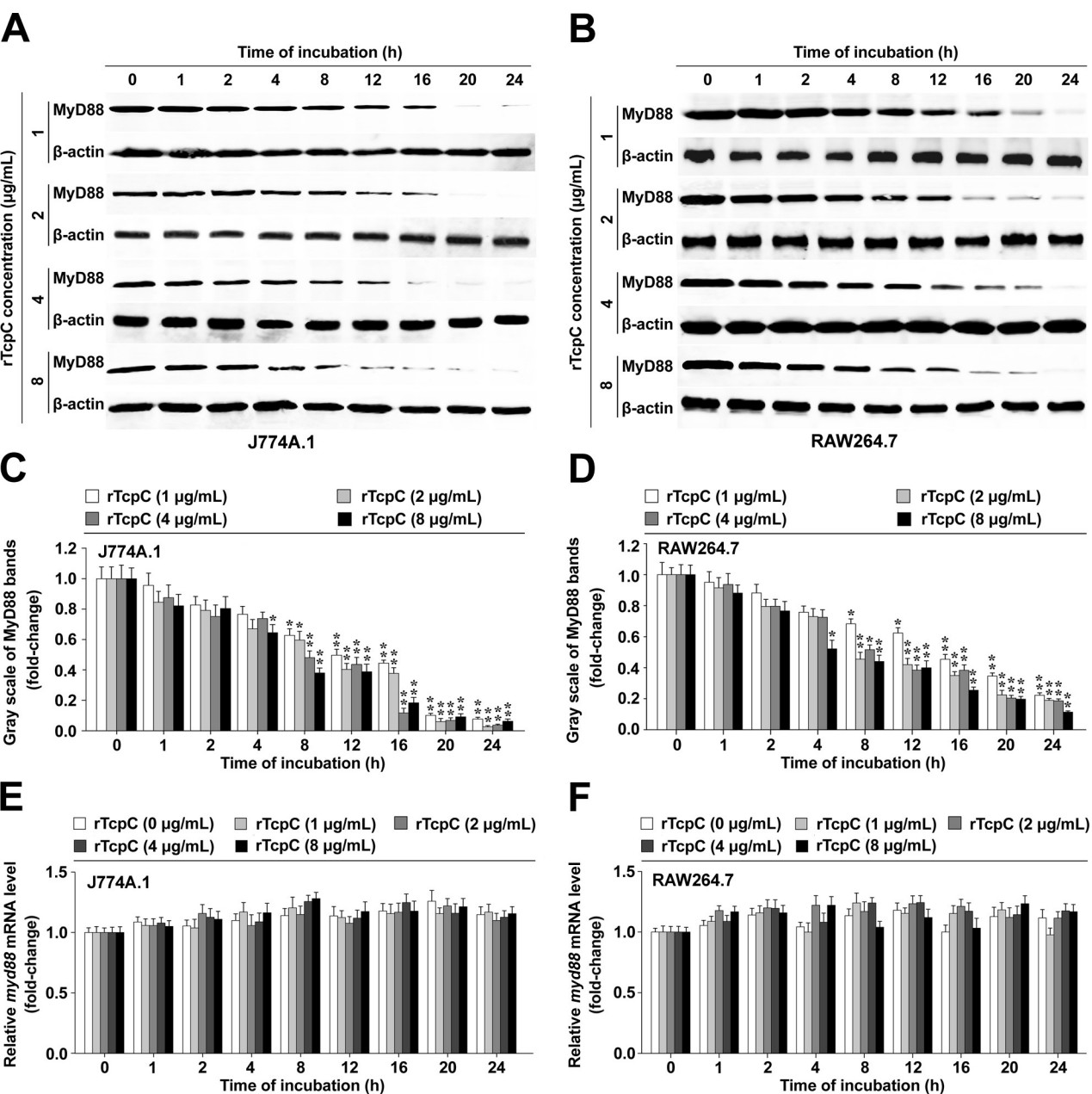

**Fig 3. rTcpC dose-dependently inhibits MyD88 protein but not mRNA in J774A.1 and RAW264.7.** (A-B) Dose-dependent inhibitory effects of rTcpC on MyD88 protein levels in J774A.1 and RAW264.7 respectively. (C-D) Gray scale analyses of MyD88 bands in different doses of rTcpC treated J774A.1 and RAW264.7. Mean ± SD of three independent experiments were shown. The MyD88 protein levels in cells without rTcpC treatment were set as 1.0. *: $p<0.05$, **: $p<0.01$ *vs* the gray scale values reflecting the MyD88 levels in cells without rTcpC treatment. (E-F) qRT-PCR detection of the influence of rTcpC on *myd88* mRNA levels in J774A.1 and RAW264.7, respectively. Mean ± SD of three independent experiments were shown. The *myd88* mRNA levels in cells without rTcpC treatment were set as 1.0.

enhanced ubiquitination could be abrogated by the E3 ubiquitin ligase inhibitor Nutlin-3 (Fig 4I and 4J). When lysates of the human macrophage cell line THP-1 were used, similar results were also obtained (S4 Fig). These data confirm that TcpC is a MyD88-targeted E3 ubiquitin ligase.

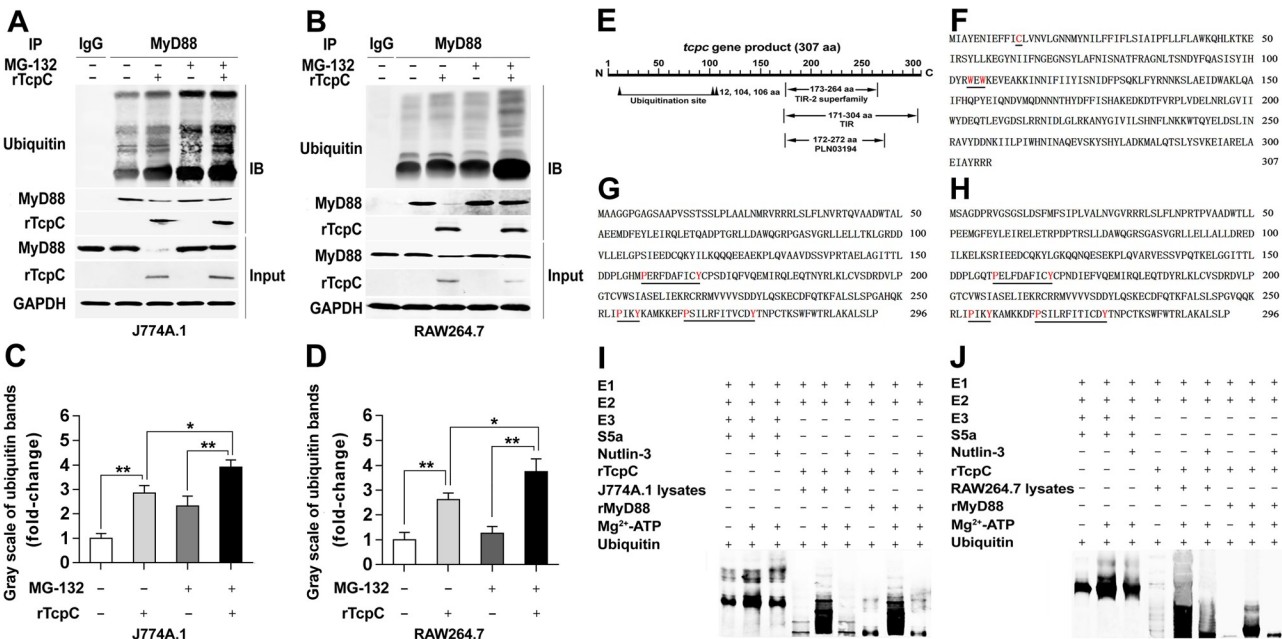

**Fig 4. rTcpC is a MyD88-targeted E3 ubiquitin ligase.** (A-B) Co-immunoprecipitation and Western blot to detect the degradation and ubiquitination of MyD88 in lysates of J774A.1 and RAW264.7, respectively. (C-D) Gray scale analyses of ubiquitination of MyD88 bands in lysates of rTcpC treated J774A.1 and RAW264.7, respectively. Mean ± SD of three independent experiments were shown. The ratio of ubiquitination to protein level of MyD88 in cells without rTcpC treatment was set as 1.0. *: $p<0.05$; **: $p<0.01$. (E) Predicted functional domains in *tcpc* gene. (F) Predicted HECT E3 ubiquitin ligase enzyme sites in amino acid sequence of TcpC. (G-H) Predicted interaction sites of HECT E3 ubiquitin ligase substrate in amino acid sequence of human and mouse MyD88, respectively. (I-J) Ubiquitination kit tests to detect the E3 activity of rTcpC. S5a, an E3 ubiquitin ligase enzyme substrate was used as the control. rTcpC was used as the E3 when lysates from J774A.1 and RAW264.7 and rMyD88 used as the substrates.

## rTcpC binds to UBE2D1 and MyD88 with high affinity

To identify the E2 that binds to TcpC, rTcpC binding proteins in the lysates of J774A.1 were co-precipitated by rTcpC-IgG. rTcpC-IgG captured four proteins from total lysates of J774A.1 (Fig 5A), and LC-MS/MS identified two of the captured proteins are MyD88 and UBE2D1 (one of the E2 enzymes) according to their cleaved peptides sequences (AVQLPKESEQNQQK and SLI-DAWQGR for MyD88, and IQKELSDLQR and HAREWTQK for UBE2D1) (Fig 5B). To determine the binding ability of rTcpC with UBE2D1 and MyD88, isothermal titration calorimetric (ITC) and surface plasmon resonance (SPR), two sensitive and reliable methods to determine protein-protein binding [31,32], were employed. The equilibrium association constant ($K_D$) values reflecting the binding ability of rTcpC with MyD88 and UBE2D1 determined by ITC were $5.39×10^{-7}$ M and $7.63×10^{-7}$ M, respectively (Fig 5C). However, when determined by SPR, the binding ability of rTcpC with UBE2D1 had a $K_D$ value of $1.72×10^{-8}$ M (Fig 5D). These data indicate that TcpC can specifically bind to UBE2D1 and MyD88 with high affinity.

## rTcpC promotes accumulation of MyD88 in proteasome

In order to further examine if rTcpC treatment leads to increased accumulation of MyD88 in proteasome, dynamic observations of co-localization of MyD88 with PSMD2 were carried out. Co-localization of MyD88 with PSMD2 in rTcpC treated J774A.1 increased initially and peaked at 4 h, then declined to form a bell-shaped curve during the 24 h treatment period (Fig 6A and 6B). The maximal co-localization percentage reached 83.6% at 4 h, and the yellow fluorescence intensity (FI) also mirrored the same trend (Fig 6B). This was probably caused by the increased degradation of MyD88 by proteasome after 4 h. Indeed, when the proteasome

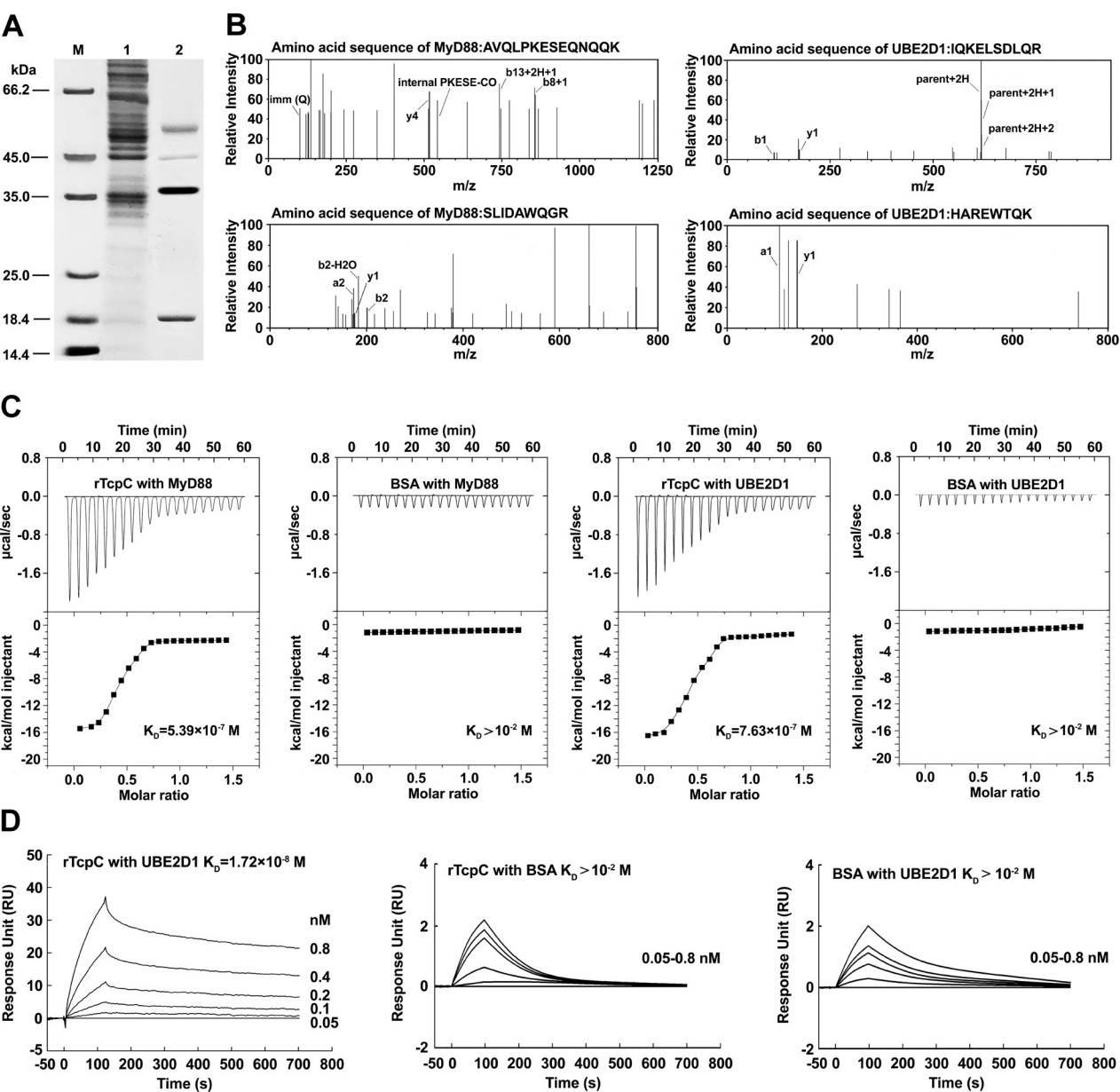

**Fig 5. rTcpC specifically binds to UBE2D1 and MyD88 with high affinity.** (A) Co-precipitation test showed the rTcpC binding proteins in J774A.1 lysates. Lane M: protein marker. Lane 1: total lysates of J774A.1. Lane 2: proteins released from protein-A-rTcpC-IgG beads. (B) LC-MS/MS analyses cleaved peptide sequences from TcpC-binding proteins. (C) ITC determination of the binding ability of rTcpC with MyD88 and UBE2D1. (D) SPR determination of the binding ability of rTcpC with UBE2D1.

activity was blocked by MG-132, the co-localization of MyD88 with PSMD2 in rTcpC treated macrophages increased along with the time during the whole 24 h treatment period (Fig 6C and 6D).

## C12 and W106 are crucial amino acids in maintaining the E3 activity of rTcpC

Previous studies reported that certain amino acid residues, such as Cys at N-terminus and WW motif, play key roles in ubiquitination by the HECT-like E3 ubiquitin ligase enzyme

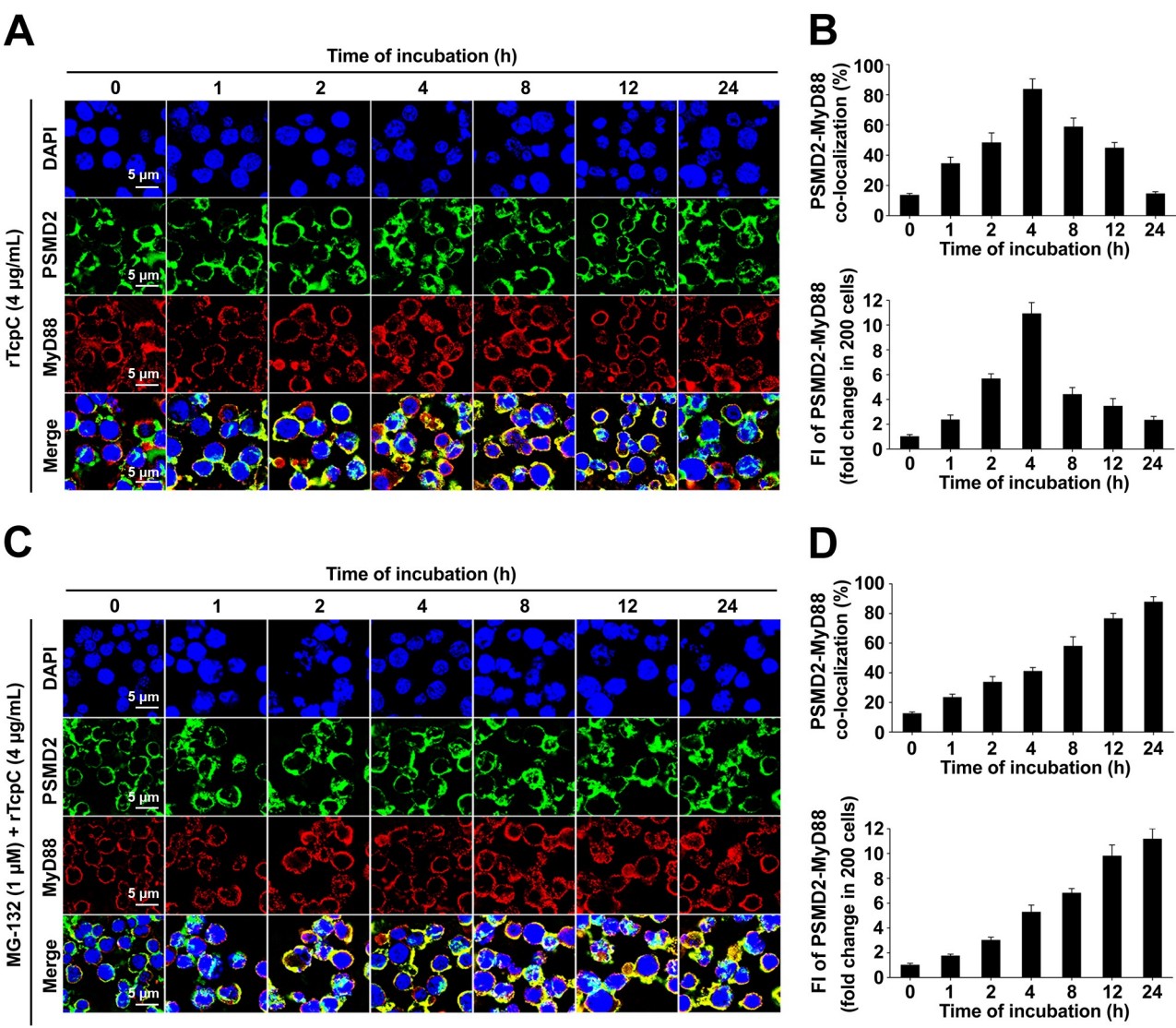

**Fig 6. TcpC promotes accumulation of MyD88 in proteasomes.** (A) Confocal microscopy to dynamically examine the co-localization of MyD88 with PSMD2 in rTcpC treated J774A.1. (B) Statistical summary of MyD88-PSMD2 co-localization percentages and FI. Mean ± SD of three independent experiments as described in A were shown. The yellow FI values from rTcpC untreated cells were set as 1.0. (C) Dynamic observation of the co-localization of MyD88 with PSMD2 under the circumstance of proteasome blockade by MG-132. (D) Statistical summary of MyD88-PSMD2 co-localization percentages and FI in experiments as described in C.

[19,25]. To identify the key amino acids that retain the E3 activity of TcpC, 4 rTcpC mutants (rTcpC-C12S, rTcpC-W104L, rTcpC-W106L and rTcpC-C12S/ W104L/W106L) were prepared by point mutations (S5A and S5B Fig), and their E3 activities were examined. Single point mutants, including rTcpC-C12S and rTcpC-W106L but not rTcpC-W104L, showed a significant decrease in E3 activity that is evidenced by their decreased ability to promote degradation (Fig 7A–7D) and ubiquitination (Fig 7E and 7F) of MyD88, compared to the rTcpC prototype. Moreover, the multiple-points mutant rTcpC-C12S/W104L/W106L could not promote degradation (Fig 7A–7D) and ubiquitination (Fig 7E and 7F) of MyD88 anymore. These data suggest that the C12 and W106 in TcpC are crucial amino acids in maintaining its E3 ligase activity.

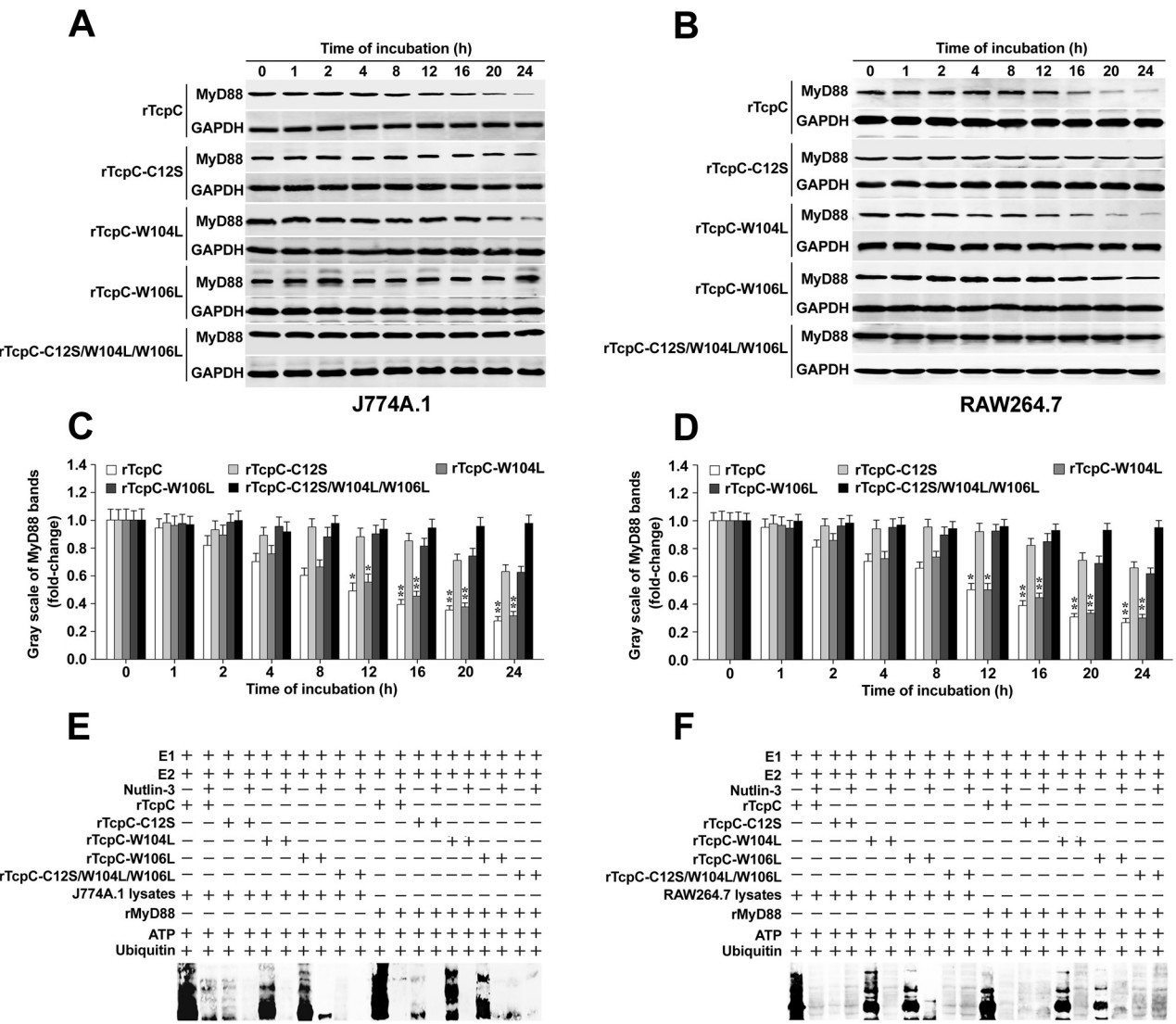

**Fig 7. C12 and W106 point mutations abrogate E3 activity of rTcpC.** (A-B) Western Blot analyses to examine the influence of rTcpC mutants on MyD88 degradation in J774A.1 and RAW264.7 respectively. (C-D) Gray scale analyses of MyD88 bands from three independent experiments as described in A-B. *: $p < 0.05$, **: $p < 0.01$ *vs* the gray scale values reflecting the MyD88 level in cells treated by the prototype rTcpC at 0 h. (E-F) Ubiquitination kit analyses to examine the influence of rTcpC mutants on ubiquitination of both proteins in lysates from J774A.1 and RAW264.7 and rMyD88.

## Discussion

*E. coli* is normally harmless commensal needs only to acquire a combination of mobile genetic elements to become a highly adapted pathogen capable of causing a range of diseases [33]. Amongst six well characterized different phylogenetic groups and nine different pathotypes of *E. coli*, UPEC is known to have the highest level of genetic and phenotypic diversity [34]. UPEC have been identified on the basis of the occurrence of genomic pathogenicity islands and the expression of virulence factors, such as adhesins, toxins, surface polysaccharides, flagella and iron-acquisition systems [1,35,36]. Unlike common pathogenic *E. coli*, UPEC colonization of the urinary tract system areas play pathogenic role during UTIs [2,37].

The innate immune system senses invading pathogens by pattern recognition receptors that include the NOD-like receptors, RIG-like receptors, cytosolic DNA receptors, and TLRs

[38]. TLRs recognize various microbial components, such as proteins, lipids, lipoproteins, nucleic acids, glycoproteins, leading to the activation of innate immune cells [39]. Upon stimulation by TLR ligands, all TLRs except TLR3 recruit the crucial adaptor protein MyD88 to the receptors [40]. Because MyD88 is the adaptor protein for all TLRs except for TLR3, that recruits the down-stream signaling molecules in TLR signaling pathway, manipulating MyD88 to evade innate immunity might be a vital strategy for bacteria to infect the host. In this respect, Cirl C, et al. have found that TcpC-TIR subverts TLR signaling and impairs the activation of the innate host defense through direct association with MyD88 [11], and Snyder GA, et al. have clarified the molecular basis of the interaction between TcpC-TIR and MyD88 [10]. But in addition to the direct association with MyD88, if there are more efficient mechanisms underlying the TcpC mediated immune evasion remains elusive. In the present study, we show, in accordance with others [11,13], that CFT073$^{wt}$ induced severe pathological changes in the kidneys, i.e. abscesses, compared to CFT073$^{\Delta tcpc}$ (Fig 1A–1D). We also observed significantly increased numbers of intracellular living bacteria and decreased levels of pro-inflammatory cytokines in CFT073$^{wt}$ treated macrophages compared with CFT073$^{\Delta tcpc}$ treatment group (Fig 2A–2D). These data demonstrate that TcpC subverts the innate immunity, leading to the enhanced pathogenicity of UPEC.

Unexpectedly, K-macrophages from CFT073$^{wt}$ infected animals and macrophages treated with CFT073$^{wt}$ showed significantly decreased protein levels of MyD88 (Figs 1E, 1F, and 2E and 2F), which was confirmed again in rTcpC treated macrophages (Figs 3A–3D, S3A, S3B, S3D and S3E), while levels of *myd88* mRNA remained unchanged (Figs 3E, 3F, S3C and S3F). Moreover, K-macrophages isolated from CFT073$^{wt}$ infected kidneys showed enhanced co-localization of MyD88 with PSMD2 (Fig 1G–1I), suggesting that TcpC may promote degradation of MyD88 through ubiquitin-proteasome pathway.

Ubiquitin-proteasome system (UPS) is the primary proteolytic route for short-lived, misfolded, and damaged proteins and has important functions in the regulation of cell signaling and transcription [41,42]. The UPS is composed of ubiquitin, E1, E2, E3, deubiquitinating enzymes, and the heart of the system, the 26S proteasome complex [19,22]. 26S proteasome is composed of 20S core particle and 19S regulatory particle [16]. Rpn1 or PSMD2 is a specific marker of the 19S subunit of proteasome and was identified as a binding site for ubiquitin chains [43]. To further confirm the degradation of MyD88 in TcpC treated macrophages was mediated by proteasomes, co-localization of MyD88 with PSMD2 was observed. Our data clearly showed that rTcpC treatment promoted accumulation of MyD88 in the proteasome (Fig 6A and 6B), which was further supported by the data that accumulation of MyD88 in proteasome increased along with the time during the 24h rTcpC treatment period when the proteasome was blocked by MG-132 (Fig 6C and 6D). These results demonstrate that TcpC promotes degradation of MyD88 through ubiquitin-proteasome pathway.

It was reported that rTcpC-TIR impaired TLR signaling and the secretion of proinflammatory cytokines in macrophages [11]. To compare rTcpC with rTcpC-TIR in the efficiency to inhibit the LPS induced production of pro-inflammatory cytokines, rTcpC-TIR was also prepared (S2A, S2C and S2D Fig), and J774A.1 and RAW264.7 were treated, in the presence or absence of LPS, with different concentrations of rTcpC and rTcpC-TIR respectively. Although 2 or 4 μg/ml of both rTcpC and rTcpC-TIR significantly inhibit LPS induced mRNA transcription and secretion of IL-1, IL-6 and TNF-α, rTcpC is more efficient than rTcpC-TIR. When the concentration was reduced to 1 μg/ml, only rTcpC but not rTcpC-TIR inhibited the LPS-induced production of inflammatory cytokines in J774A.1 and RAW264.7 (S6A and S6B Fig). These data demonstrate that the full length rTcpC protein is more potent than the rTcpC-TIR to inhibit TLR signaling.

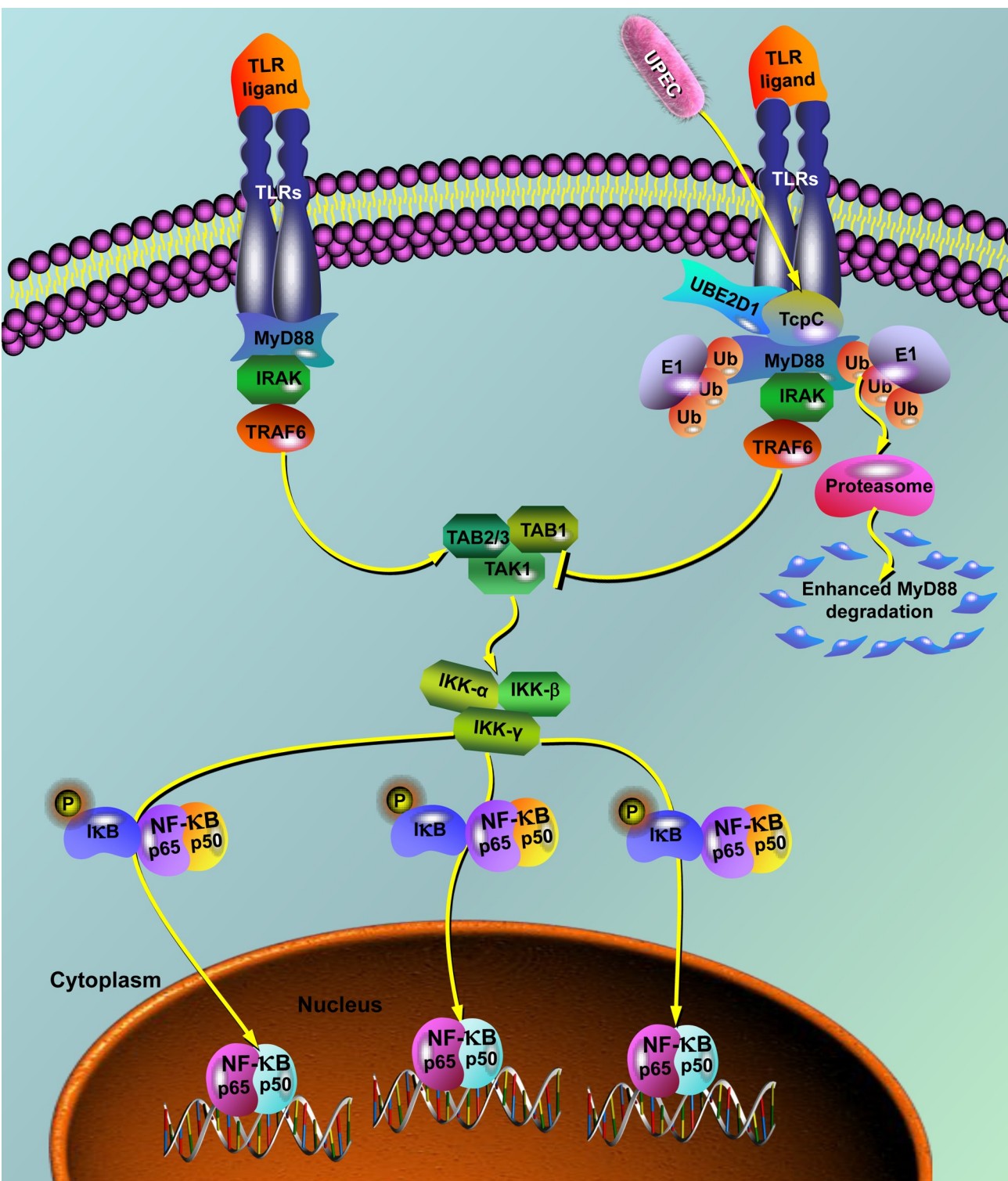

**Fig 8. Schematic diagram of the molecular mechanisms by which TcpC blocks TLR signaling pathway.** On stimulation by TLR ligand, all TLRs except for TLR3 recruit MyD88 via interactions between TIR domains on each other. MyD88 then recruit the down-stream signal molecules, leading to the activation of NF-κB. Under the circumstance of UPEC infection, TcpC blocks TLR signaling pathway though two mechanisms: 1) TcpC-TIR blocks the interaction between TLR and MyD88 by direct association with MyD88; 2) TcpC functions as an E3 ubiquitin ligase, in combination with UBE2D1, to promote ubiquitination of MyD88, leading to degradation of MyD88 by proteasomes.

**Table 1. Primers used in this study.**

| Primer | Sequence (5' to 3') | Purpose |
|---|---|---|
| *MyD88* | F: GGCTGCTCTCAACATGCGA | Detection of *MyD88* mRNA |
| | R: CTGTGTCCGCACGTTCAAGA | |
| *IL-1β* | F: CCTTGTGCAAGTGTCTGAAG | Detection of *IL-1β* mRNA |
| | R: GGGCTTGGAAGCAATCCTTA | |
| *IL-6* | F: GCCCTTCAGGAACAGCTATGA | Detection of *IL-6* mRNA |
| | R: TGTCAACAACATCAGTCCCAAGA | |
| *TNF-α* | F: CCTGTAGCCCACGTCGTAG | Detection of *TNF-α* mRNA |
| | R: GGGAGTAGACAAGGTACAACCC | |
| *β-actin* | F: ATGGATGACGATATCGCTG | Inner reference used in qRT-PCRs |
| | R: AACACCCATTCCCTTCACAG | |

F: forward primer. R: reverse primer.

Although a number of E3 ubiquitin ligases from bacterial pathogens were reported, such as AvrPtoB [44], NleG [45], SopA [46,47], NleL [48], IpaH [49], IpaH3 [50], SspH2 [51], the studies mainly focused on their structures and their precise biochemical activities have been elucidated in very few cases. Furthermore, no bacterial pathogen derived E3 ubiquitin ligase targeting proteins in TLR signaling pathway was reported.

Taken together, in the present study we show for the first time that TcpC is a MyD88-targeted E3 ubiquitin ligase that promotes degradation of MyD88 through ubiquitin-proteasome pathway, hereby blocking the TLR signaling pathway (Fig 8). To our knowledge, this is also the first example that bacterial pathogens inhibit MyD88-mediated signaling pathway by virulence factors that function as E3 ubiquitin ligase. It is obvious that manipulating MyD88 by enzyme-mediated degradation to interrupt the TLR signaling pathway is a more efficient way for UPEC to evade innate immunity, which is reasonable in the evolution of pathogenic microbes. Our findings not only illuminate the biochemical mechanisms underlying the TcpC mediated immune evasion, but also provide novel clues to clarify the pathogenicity of other bacterial pathogens.

Since ubiquitination plays important role in antigen processing, autophagy and immune cell differentiation, the regulatory effects of TcpC on differentiation and function of dendritic cells, autophagy and polarization of macrophages deserve further investigation. Furthermore, the characteristic histological changes of UPEC infected acute pyelonephritis are neutrophil infiltration, influence of TcpC on the function of neutrophils also needs to be explored.

## Materials and methods

### Ethics statement

Animal experiments were performed in accordance with the National Regulations for the Administration of Experimental Animals of China (1988–002) and the National Guidelines for Experimental Animal Welfare of China (2006–398). All the animal experimental protocols were approved by the Ethics Committee for Animal Experiment of Zhejiang University City College School of Medicine.

### Uropathogenic *E. coli* strain and culture

TcpC-secreting CFT073[wt] was kindly provided by Professor Jian-guo Xu (State Key Laboratory for Infectious Disease Prevention and Control, National Institute for Communicable

Disease Control and Prevention, China). CFT073$^{\Delta tcpc}$ was constructed in our laboratory as described previously [8]. The strains were cultured in LB medium at 37˚C.

## Cell lines and culture

The mouse macrophage cell lines J774A.1 (ATCC TIB-67), RAW264.7 (ATCC TIB-71) and human macrophage cell line THP-1 (ATCC TIB-202) were obtained from Shanghai Institute of Biochemistry and Cell Biology (Shanghai, China). The cells were grown in RPMI-1640 medium containing 10% fetal calf serum (FBS) (Gibco, USA) plus 100 U/ml penicillin and 100 μg/ml streptomycin (Sigma, USA) in a humidified atmosphere of 5% $CO_2$ at 37˚C.

## Mouse pyelonephritis model

Female C57BL/6 mice, 6–8 weeks of age, were purchased from SLC Laboratory Animal Co. Ltd. (Shanghai, China) and were housed in specific pathogen-free conditions. Mouse pyelonephritis models were prepared as previously described [8]. In briefly, mice were anesthetized with Avertin (40 mg of 2,2,2-tribromoethanol (Sigma, USA) dissolved in 1 ml of tertamyl alcohol (0.01 ml/g body weight intraperitoneally)) and infected by transurethral instillation of $10^8$ CFU of CFT073$^{wt}$ or CFT073$^{\Delta tcpc}$ using a flexible polyethylene catheter. Mice treated with normal saline served as the control group. The mice were sacrificed 3 days later and the kidneys were obtained for pathological examination as described previously [8].

## Western blotting analyses of MyD88 and NF-κB

Proteins from lysates of following three different treatment macrophages were prepared as described in our previous reports [8,52]. 1) K-macrophages isolated from murine models with pyelonephritis; 2) To detect the influence of TcpC on protein levels of MyD88 and NF-κB, $5×10^5$ cells of BMDM, J774A.1 or RAW264.7 were separately co-cultured in transwell with CFT073$^{wt}$ or CFT073$^{\Delta tcpc}$ at MOI = 100 for the indicated time; 3) To detect the influence of rTcpC or rTcpC mutant on the protein levels of MyD88, $5×10^5$ cells of J774A.1, RAW264.7, THP-1 or BMDM were cultured in 6-well culture plates (Corning, USA) overnight. Then cells were treated with different concentrations of rTcpC (1, 2, 4 or 8 μg/ml) or 4 μg/ml of rTcpC or rTcpC mutant for the indicated time. Proteins were submitted to SDS-PAGE and electro-transferred onto PVDF membrane (Bio-Rad, USA). MyD88 was probed using rabbit MyD88-IgG (Cell Signaling Technology, USA) as the primary antibody. NF-κB p50, p65, p-p50 and p-p65 were probed by rabbit NF-κB p105/p50-IgG, rabbit NF-κB p65-IgG, rabbit phospho-NF-κB p105 (Ser933)-IgG and rabbit phospho-NF-κB p65 (Ser536)-IgG (Cell Signaling, USA) as the primary antibody respectively. IRDye 680RD goat anti-rabbit-IgG (H+L) (LI-COR, USA) was used as the secondary antibody. The images were developed on the ODYSSEY CLx Infrared Imaging System (LI-COR). Gray scale analyses of bands were made from data of at least three independent experiments. β-actin or GAPDH were used as the inner references. BMDM were prepared as described previously [11].

## Confocal microscopy to detect co-localization of MyD88 with PSMD2

$5×10^5$ cells of K-macrophages from different groups and 4 μg/ml rTcpC treated J774A.1 cells were centrifuged at 800×g for 10 min at 4˚C. Mouse anti-MyD88-IgG (Abcam) or rabbit anti-PSMD2-IgG (Abcam) were used as the primary antibody. Alexa Fluor 488-conjugated goat anti-rabbit-IgG and Alexa Fluor 647-conjugated goat anti-mouse-IgG (Abcam) were used as the secondary antibody respectively and DAPI (Sigma) to stain the nucleus. Co-localization of MyD88 with PSMD2 was examined by confocal microscopy (Olympus) (495/519, 652/668 or

485 nm excitation/emission wavelengths for Alexa Fluor 488, Alexa Fluor 647 or DAPI detection). Percentages and yellow fluorescence intensities (FI) reflecting the co-localization were analyzed. Untreated corresponding cells were used as controls. To examine the influence of MG-132 on rTcpC induced accumulation of MyD88 in proteasomes, J774A.1 cells were treated with 1 μM MG-132 for 30 minutes before treatment with 4 μg/ml rTcpC for the indicated time, and then the treated cells are subjected to confocal microscopy as described above.

## Detection of viability of UPEC from infected macrophages

$5 \times 10^5$ cells of BMDM, J774A.1 and RAW264.7 were co-cultured with CFT073$^{wt}$ or CFT073$^{\Delta tcpc}$ at MOI = 100 for 12 h. After washing with PBS and trypsinization, the co-cultures were collected and centrifuged at 500×g for 5 min (4°C). After incubation with 100 U/ml penicillin and 100 μg/ml streptomycin for 30 min, the cells were washed with PBS three times (500×g for 5 min each time) to remove extracellular bacteria in the supernatants and then lysed with 0.05% NaTDC-PBS to release intracellular bacteria. After a 5 min centrifugation at 500×g (4°C) to remove cell debris, the lysates were centrifuged at a 12,000×g centrifugation at 4°C for 30 min to precipitate intracellular bacteria. Subsequently, the viability of bacteria was detected by confocal microscopy (Olympus, Japan) (485/630 nm excitation/emission wavelengths for SYTO 9) using a LIVE/DEAD Bacterial Viability Kit (Thermo) [53]. CFT073$^{wt}$ or CFT073$^{\Delta tcpc}$ cultured in LB medium were used as the control groups.

## qRT-PCR

To determine the influence of TcpC on mRNA expression of *IL-1β*, *IL-6* and *TNF-α*, $5 \times 10^5$ cells of BMDM, J774A.1 and RAW264.7 were separately co-cultured in transwell with CFT073$^{wt}$ or CFT073$^{\Delta tcpc}$ at MOI = 100 for 16 h. To test the influence of rTcpC or rTcpC-TIR on mRNA expression of *IL-1β*, *IL-6* and *TNF-α*, $5 \times 10^5$ cells of J774A.1 and RAW264.7 were treated with different concentrations (1, 2 or 4 μg/ml) of rTcpC or rTcpC-TIR for 16 h. To detect the influence of rTcpC on mRNA expression of *myd88*, $5 \times 10^5$ cells per well of J774A.1, RAW264.7, THP-1 or BMDM were cultured in 6-well culture plates (Corning, USA) overnight. Then the cells were treated with different concentrations of rTcpC (1, 2, 4 or 8 μg/ml) for 1, 2, 4, 8, 12, 16, 20 or 24 h respectively. RNAs were extracted and used for qRT-PCR to detect mRNA levels as previously described [52]. Data were analyzed using the ΔΔCt model and randomization test in REST 2005 software [52,54]. The primers used in qRT-PCR are listed in Table 1.

## ELISA

Culture supernatants of cells treated as described in qRT-PCR were harvested and used for measurement of cytokine levels by ELISA kits (eBioscience, USA) according the instructions of the manufacturer.

## Co-immunoprecipitation and immunoblotting to detect ubiquitination of MyD88

$5 \times 10^5$ cells of J774A.1 and RAW264.7 were treated with 4 μg/ml rTcpC for 16 h and collected by centrifugation (800×g for 10 min at 4°C). The cells were lysed with RIPA lysis buffer (Beyotime BioTech). The lysates were centrifuged at 12000×g for 30 min to remove cell debris. Protein concentration in the supernatants was determined by a BCA Protein Assay Kit (Thermo Scientific). Proteins in cell lysates were then immunoprecipitated (IP) with the anti-MyD88 IgG (Cell Signaling) antibody using Pierce Protein A/G Magnetic Beads (Thermo Scientific)

and the retrieved proteins (rTcpC, MyD88 and ubiquitin) were detected by immunoblotting (IB) with corresponding antibodies as described above and previously [8,19].

## Bioinformatics analyses

The structures and functional domains in TcpC, MyD88 of human and mouse were analyzed using NCBI-Batch CD-Search software [55].

## *In vitro* ubiquitination kit assays

Ubiquitination assays were performed according to the instruction of the manufacturer (Boston Biochem, USA). Briefly, the following components were successively added into 0.5 ml polypropylene tubes on ice: 9 μl dH$_2$O, 3 μl 10× reaction buffer, 3 μl 10× rS5a, cell lysates or rMyD88 substrate protein, 3 μl 10× E1 enzyme, 3 μl 10× E2 enzyme, 3 μl 10× MuRF1 enzyme or rTcpC, 3 μl 10× Mg$^{2+}$-ATP solution. Reactions were incubated at 37°C for 1 h and the products were subjected to Western Blot analysis [19]. To determine the MyD88-specific E3 activity of rTcpC, E3 and its substrate S5a in the kit were replaced by rTcpC and lysates of macrophages or rMyD88 (Abcam, USA), respectively.

## Co-precipitation assay to pull down rTcpC-binding proteins in macrophages

Freshly-cultured J774A.1 cells were washed twice with PBS and were harvested by centrifugation. The cells were suspended in PBS and then lysed with RIPA lysis buffer (Beyotime Bio-Tech). The lysates were centrifuged at 12,000×g for 30 min at 4°C to remove cell debris. The supernatants were collected and protein concentrations were determined by a BCA Protein Assay Kit (Thermo Scientific, USA). 20 μg rabbit anti-rTcpC-IgG in 500 μl PBS were mixed with 100 μl of 6 mg/ml protein-A-coated agarose beads (Millipore, USA), and incubated in a 90 rpm rotator at 4°C overnight to form protein-A-rTcpC-IgG beads. After a 10-min centrifugation (14,000×g at 4°C) and 3 times wash with PBS, the beads were suspended in 500 μl PBS and then incubated with 20 μg rTcpC to form protein-A-rTcpC-IgG-rTcpC beads. After centrifugation and wash with PBS, the beads were suspended in 500 μl PBS and then incubated, at 4°C for 2 h, with 200 μg proteins in the lysates in a 90 rpm rotator. After centrifugation and wash thoroughly with PBS, the beads were suspended in Laemmili SDS-PAGE sample buffer for a 5-min water-bath at 100°C to release protein-A-rTcpC-IgG-binding proteins. After a 10-min centrifugation at 14,000×g (4°C), the retrieved proteins were subjected to SDS-PAGE.

## LC-/MS/MS to identify rTcpC-binding proteins in J774A.1

Protein bands on the above SDS-PAGE were identified by liquid chromatography plus a type LC1000-LTQ tandem mass spectrometry (LC-MS/MS, Thermo Scientific). The obtained data were automatically searched against the protein database using Proteome Discoverer 1.4 software.

## Determination of binding abilities of rTcpC with UBE2D1 and MyD88

Binding abilities of rTcpC with UBE2D1 and MyD88 were determined by SPR and ITC [31,32]. Briefly, for SPR detection, 1 nM UBE2D1 (R&D, USA) was cross-linked on the activated CM5 sensing array (GE, USA) and then 0.05–0.8 nM rTcpC in PBS flowed through the surface of UBE2D1-binding array. The combination of rTcpC with UBE2D1 was detected using a SPR-based detector (Type-T200, GE) and quantified by the values of equilibrium association constant (K$_D$). For ITC detection, 1 μM rTcpC in PBS was added in the titration pool

while 0.1 μM UBE2D1 or rMyD88 in PBS were added in the sample pool. The $K_D$ values reflecting the combination of rTcpC with UBE2D1 or rMyD88 in titrating process were detected using a type VP-ITC microcalorimeter (MicroCal, USA) and then analyzed using Origin software [52]. BSA (Sigma, USA) was used as the negative array-linking fixed and mobile phase controls in SPR and the negative titration control in ITC.

## Functional determination of rTcpC mutants

The single point rTcpC mutants (rTcpC-C12S, rTcpC-W104L and rTcpC-W106L) and multiple-points rTcpC mutant (rTcpC-C12S/W104L/W106L) were prepared by recombinant techniques as described in Supplemental Information. The primers used in construction of rTcpC mutants are listed in S1 Table. The influence of these mutants on MyD88 protein levels in treated macrophages and their E3 ubiquitin ligase activities were determined by Western Blot and *in vitro* ubiquitination kit assay as described above.

## Statistical analyses

Data from a minimum of three independent experiments were averaged and presented as mean ± standard deviation (SD). Mann-Whitney followed by Dunnett's multiple comparisons test was used to determine significant differences. Statistical significance was defined as $p < 0.05$, extremely significance was defined as $p < 0.01$.

## Raw data

Numerical values were used to generate histograms and all blots in manuscript are in S1 Histograms data and S1 Raw data blots.

## Supporting information

**S1 Fig. Purity identification of K-macrophages and BMDM by FACS.** (A) CD11b and F4/80 expressions in K-macrophages isolated from control and PN mice models. (B) CD11b and F4/80 expressions in BMDM.
(TIF)

**S2 Fig. rTcpC and rTcpC-TIR expression and purification.** (A) Expression and purification of rTcpC and rTcpC-TIR. Lane M: protein marker. Lane 1: pET42a transformed *E. coli* BL21DE3. Lane 2: pET42a-*tcpc* transformed *E. coli* strain BL21DE3. Lane 3: Purified rTcpC. Lane 4: pET42a-*tcpc-tir* transformed *E. coli* strain BL21DE3. Lane 5: Purified rTcpC-TIR. (B) Elution curve of rTcpC. (C) Elution curve of rTcpC-TIR. (D) Detection of LPS in rTcpC or rTcpC-TIR preparation.
(TIF)

**S3 Fig. rTcpC inhibits MyD88 protein levels in THP-1 and BMDM.** (A) Dose-dependent inhibitory effects of rTcpC on MyD88 protein levels in THP-1. (B) Gray scale analyses of MyD88 bands in THP-1 treated with different doses of rTcpC. Mean ± SD of three independent experiments were shown. The MyD88 protein levels in cells without rTcpC treatment were set as 1.0. *: $p < 0.05$, **: $p < 0.01$ *vs* the gray scale values reflecting the MyD88 levels in cells without rTcpC treatment. (C) Dynamic analyses, by qRT-PCR, of the influence of rTcpC on *myd88* mRNA levels in THP-1. Mean ± SD of three independent experiments were shown. The *myd88* mRNA levels in cells without rTcpC treatment were set as 1.0. (D) Inhibitory effects of rTcpC on MyD88 protein levels in BMDM. (E) Gray scale analyses of MyD88 bands in experiments as described in D. Mean ± SD of three independent experiments were shown.

The MyD88 protein levels in cells without rTcpC treatment were set as 1.0. *: $p<0.05$, **: $p<0.01$ *vs* the gray scale values reflecting the MyD88 levels in cells without rTcpC treatment. (F) Dynamic analyses of the influence of rTcpC on *myd88* mRNA levels in BMDM.
(TIF)

**S4 Fig. rTcpC enhance ubiquitination of proteins in THP-1 lysates.** Ubiquitination kit tests to detect the E3 activity of rTcpC. S5a, an E3 ubiquitin ligase enzyme substrate was used as the control. rTcpC was used as the E3 when lysates from THP-1 and rMyD88 used as the substrates.
(TIF)

**S5 Fig. rTcpC mutants expression and purification.** (A) SDS-PAGE analyses of expression and purification of rTcpC mutants. Upper: Expression of rTcpC mutants. Lane M: Protein marker. Lane 1: pET42a transformed *E. coli* BL21DE3. Lanes 2–5: pET42a-*tcpc-C12S-*, pET42a-*tcpc-W104L-*, pET42a-*tcpc-W106L-* and pET42a-*tcpc-C12S/W104L/W106L-*transformed *E. coli* BL21DE3, respectively. Bottom: Purity identification of rTcpC mutants. Lane M: Protein marker. Lane 1–4: rTcpC-C12S, rTcpC-W104L, rTcpC-W106L and rTcpC-C12S/W104L/W106L, respectively. (B) Detection of LPS in rTcpC mutants by spectrophotometric limulus test.
(TIF)

**S6 Fig. rTcpC is more potent than rTcpC-TIR to inhibit LPS induced expression of proinflammatory cytokines.** (A) qRT-PCR to examine the influence of rTcpC and rTcpC-TIR on mRNA levels of *IL-1β*, *IL-6* and *TNF-α*. *: $p<0.05$, **: $p<0.01$, NS: no sense. (B) ELISA to detect protein levels of IL-1β, IL-6 and TNF-α. *: $p<0.05$, **: $p<0.01$, NS: no sense.
(TIF)

**S1 Table. The primers used in construction of rTcpC and rTcpC mutants.**
(DOCX)

**S1 Histograms data. Numerical values were used to generate histograms in manuscript.**
(RAR)

**S1 Raw data blots. Raw data blots in manuscript.**
(RAR)

## Acknowledgments

We thank Professor Jian-guo Xu (State Key Laboratory for Infectious Disease Prevention and Control, National Institute for Communicable Disease Control and Prevention, China) for providing the UPEC strain CFT073.

## Author Contributions

**Conceptualization:** Jia-qi Fang, Thomas Miethke, Jian-ping Pan.

**Data curation:** Jia-qi Fang, Qian Ou, Jun Pan, Jie Fang, Da-yong Zhang, Miao-qi Qiu, Yue-qi Li, Xiao-Hui Wang, Xue-yu Yang, Zhe Chi, Wei Gao, Jun-ping Guo, Jian-ping Pan.

**Formal analysis:** Jia-qi Fang, Qian Ou, Jun Pan, Jian-ping Pan.

**Writing – original draft:** Jia-qi Fang, Thomas Miethke, Jian-ping Pan.

**Writing – review & editing:** Jia-qi Fang, Jian-ping Pan.

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
