## [Decision Letter · Decision Letter 0]

21 Nov 2020

Dear Dr. Fang,

Thank you very much for submitting your manuscript "TcpC inhibits toll-like receptor signaling pathway by serving as an E3 ubiquitin ligase that promotes degradation of myeloid differentiation factor 88" for consideration at PLOS Pathogens. As with all papers reviewed by the journal, your manuscript was reviewed by members of the editorial board and by several independent reviewers. The reviewers appreciated the attention to an important topic. Based on the reviews, we are likely to accept this manuscript for publication, providing that you modify the manuscript according to the review recommendations. Comments by Reviewers 1 and 2 are provided here as attachments. All reviewers pointed out the need for some additional controls and better explanations of what the controls are. Please ensure that the correct statistical approaches are used and consider adjusting the Discussion to place the new findings in the broader context of the field rather than just reiterating the results. 

Sincerely,

Matthew A Mulvey, Ph.D.

Associate Editor

PLOS Pathogens

Guy Tran Van Nhieu

Section Editor

PLOS Pathogens

Kasturi Haldar

Editor-in-Chief

PLOS Pathogens

orcid.org/0000-0001-5065-158X

Michael Malim

Editor-in-Chief

PLOS Pathogens

orcid.org/0000-0002-7699-2064

Reviewer Comments (if any, and for reference):

Reviewer's Responses to Questions

**Part I - Summary**

Reviewer #1: (No Response)

Reviewer #3: The paper by Fang et al. presents interesting and novel data regarding the mechanism through which TcpC affects the host immune response via MyD88 protein.

The authors use a tcpC mutant strain compared to a wild-type strain to demonstrate the importance of the protein for virulence/colonization during UTI in a mouse model as well as entry/survival in macrophage and macrophage cell lines of murine origin. Globally, the study is well conducted and experiments well planned. The text is also clearly organized and written.

Strengths of the work include the combination of methods used to demonstrate the activity of the TcpC protein to act as a ubiquitin ligase and decrease levels of MyD88 protein as well as demonstration of key roles of specific residues for this activity, as well as demonstration of the partner proteins identified by mass spectrometry.

A drawback of the study concerns use of only mouse derived cells and protein although it is likely that this bacterial protein acts similarly on human host targets to abate the innate immune response.

**Part II – Major Issues: Key Experiments Required for Acceptance**

Reviewer #1: (No Response)

Reviewer #3: Some specific comments/critiques follow:

1. For studies concerning macrophages or macrophage cell lines, all cell types tested are only murine source. Why were only mouse cell lines used? It would be worthwhile determining if the same responses occur with human derived macrophage cell lines or primary human macrophages.It would be of interest to confirm the same mechanism in a human source cell or cell line or human derived MyD88 protein sequence.

Fig. 1. There seems to be a discoloration of the kidney with a paleness with WT CFT073, and an important increase in redness in the tcpC mutant infected kidneys. In B vs C is there a cellular content present that correlates with the change in color observed in the kideys? The term cellular infiltrates is vague and some of the basic changes in cell content could be described. More or less types of what cells??

Fig. 1C and D. The change in level of MyD88 detected seems to be potentially modest when considering differences in cellular composition in the tissues. This is an important aspect to consider from the wt vs. delta tcpC mutant infected kidneys and even the control samples (as seen in Fig. 1B). The enrichment used to obtain K-macrophages was cell sorting with the CD11b marker, which is not specific to macrophages, but common to most phagocytic and other cells (monocytes, neutrophils, macrophages, NK and even other cell types). To what extent can the differences in MyD88 quantification be attributed to cell composition versus differences in potential degradation or reduced levels of MyD88 in actual K-Macrophages when considering the Western blot results?

**Part III – Minor Issues: Editorial and Data Presentation Modifications**

Reviewer #1: (No Response)

Reviewer #2: (No Response)

Reviewer #3: Some recommendations to improve the manuscript :

1. Pathogenic E. coli causing extra-intestinal infections such as UTIs and sepsis represent a diverse group of isolates tat can contain a variety of virulence factors. This work focuses on a protein from a well studied from from a case of pyelonephritis. It would be worthwhile for the authors to also present a few sentences concerning the prevalence of TcpC among different ExPEC UPEC strains and among specific phylogenetic groups, sequence types, etc. There are quite a few publications describing the frequency of this virulence factor and its association with particular subgroups of E. coli. This would better substantiate the potential general role of TcpC for UTIs and other systemic infections cause by E. coli.

In Fig. 2a, it is unclear what the control shown is : the labeling indicates macrophages or cells lines, but is the control just bacteria that were not interacting with macrophage cells? It is also not clear why bacterial cell enumeration by viable counts after macrophage cell lysis was not determined as a direct method to quantify total differences in survival during macrophage interaction.

Fig. 2. For plates G and H, the time of the sampling after infection should be indicated in the Figure legend.

Fig. 3. The names of the cell lines tested should be labeled directly in the figures for more clarity. It would have been of interest to determine if a similar decrease in MyD88 occured in primary macrophages as well.

Fig. 7. Write the name of the cells as a label title above the Plates at top of Fig. 7, since J774 corresponds to A, C, E and RAW to B, D, F

5. The discussion is limited in discussing the results in relation to other studies and is mainly presented as a revisitating of the data, making it rather repetitive. Without referring or comparing results or aspects to other studies, the discussion is lacking scope or comparison of other aspects related to discoveries presented, such as the existence of other bacterial ubiquitin ligases that play a role in virulence through different pathways for example. The paragraphs in the discussion are mainly repetitive descriptions of the results only, and need to be discussed in contrast or comparison to other reports or literature.

Supplementary data:

Why are there no legends/text to explain what is shown in the plates in the supplemental figures? This should be independently clear without needing to refer to any text in nother section or the main manuscript. Without legends, it is not possible to determine what is being presented.

PLOS authors have the option to publish the peer review history of their article (what does this mean?). If published, this will include your full peer review and any attached files.

Reviewer #1: No

Reviewer #2: No

Reviewer #3: No
---

## [Editor Report · Decision Letter 1]

9 Feb 2021

Dear Dr. Fang,

Thank you very much for submitting your manuscript "TcpC inhibits toll-like receptor signaling pathway by serving as an E3 ubiquitin ligase that promotes degradation of myeloid differentiation factor 88" for consideration at PLOS Pathogens. We appreciate your careful responses to the previous critiques and feel that the paper is much improved and will be an important addition to the field. However, we have a couple of concerns that we feel should be addressed more carefully before we can accept the manuscript. 

1. Previous reviewers questioned the use of LAMP1 as "one of the generally accepted markers of the proteasome". While LAMP1 is often used as a marker of lysosomes and occasionally autophagosomes, it is not clear that this is a commonly used proteasome marker. Please provide references for use of LAMP1 as a proteasome marker, or otherwise clarify this issue. For example, co-localization of MyD88 with proteasomes in a single cell line ± TcpC treatment using antibody specific for one of the proteasome subunits would be reassuring. 

2. With the addition of comments on other E3 ligases, the Discussion is much improved. However, the majority of the Discussion is still a re-hashing of the Results section. It might be helpful to readers if you could place your findings in a broader context, perhaps by briefly considering future directions and new questions presented by your work.

We are likely to accept this manuscript for publication, providing that you modify the manuscript according to the  recommendations. 

Sincerely,

Matthew A Mulvey, Ph.D.

Associate Editor

PLOS Pathogens

Guy Tran Van Nhieu

Section Editor

PLOS Pathogens

Kasturi Haldar

Editor-in-Chief

PLOS Pathogens

orcid.org/0000-0001-5065-158X

Michael Malim

Editor-in-Chief

PLOS Pathogens

orcid.org/0000-0002-7699-2064
---

## [Editor Report · Decision Letter 2]

17 Mar 2021

Dear Dr. Fang,

We are pleased to inform you that your manuscript 'TcpC inhibits toll-like receptor signaling pathway by serving as an E3 ubiquitin ligase that promotes degradation of myeloid differentiation factor 88' has been provisionally accepted for publication in PLOS Pathogens.

Best regards,

Matthew A Mulvey, Ph.D.

Associate Editor

PLOS Pathogens

Guy Tran Van Nhieu

Section Editor

PLOS Pathogens

Kasturi Haldar

Editor-in-Chief

PLOS Pathogens

orcid.org/0000-0001-5065-158X

Michael Malim

Editor-in-Chief

PLOS Pathogens

orcid.org/0000-0002-7699-2064

Reviewer Comments (if any, and for reference): a few minor grammatical issues

---

## [Editor Report · Acceptance letter]

26 Mar 2021

Dear Dr Pan,

We are delighted to inform you that your manuscript, "TcpC inhibits toll-like receptor signaling pathway by serving as an E3 ubiquitin ligase that promotes degradation of myeloid differentiation factor 88," has been formally accepted for publication in PLOS Pathogens.

Best regards,

Kasturi Haldar

Editor-in-Chief

PLOS Pathogens

orcid.org/0000-0001-5065-158X

Michael Malim

Editor-in-Chief

PLOS Pathogens

orcid.org/0000-0002-7699-2064